# Pharmacogenetics of Neoadjuvant MAP Chemotherapy in Localized Osteosarcoma: A Study Based on Data from the GEIS-33 Protocol

**DOI:** 10.3390/pharmaceutics16121585

**Published:** 2024-12-12

**Authors:** Juliana Salazar, María J. Arranz, Javier Martin-Broto, Francisco Bautista, Jerónimo Martínez-García, Javier Martínez-Trufero, Yolanda Vidal-Insua, Aizpea Echebarria-Barona, Roberto Díaz-Beveridge, Claudia Valverde, Pablo Luna, María A. Vaz-Salgado, Pilar Blay, Rosa Álvarez, Ana Sebio

**Affiliations:** 1Translational Medical Oncology Laboratory, Institut de Recerca Sant Pau (IR Sant Pau), 08041 Barcelona, Spain; 2Research Laboratory Unit, Fundació Docència i Recerca Mútua Terrassa, 08221 Terrassa, Spain; mjarranz@mutuaterrassa.es; 3Medical Oncology Department, Hospital Universitario Fundación Jiménez Díaz, 28040 Madrid, Spain; jmartin@atbsarc.org; 4Pediatric Hematology and Oncology Department, Hospital Niño Jesús, 28009 Madrid, Spain; f.j.bautistasirvent@prinsesmaximacentrum.nl; 5Princess Maxima Centrum for Pediatric Cancer, 3584 CS Utrecht, The Netherlands; 6Medical Oncology Department, Hospital Universitario Virgen de la Arrixaca, 30120 El Palmar, Spain; jeronimo@seom.org; 7Medical Oncology Department, University Hospital Miguel Servet, 50009 Zaragoza, Spain; jmtrufero@seom.org; 8Medical Oncology Department, Complejo Hospitalario Universitario de Santiago de Compostela, 15706 Santiago de Compostela, Spain; yolanda.vidal.insua@sergas.es; 9Pediatric Oncology Group, Pediatrics Department, Hospital Universitario Cruces, 48940 Barakaldo, Spain; aizpeabeatriz.echebarriabarona@osakidetza.eus; 10Medical Oncology Department, Hospital Universitario y Politécnico La Fe de Valencia, 46026 Valencia, Spain; robertdiazbeveridge@gmail.com; 11Medical Oncology Department, Hospital Universitari Vall d’Hebrón and Vall d´Hebrón Institute of Oncology (VHIO), 08035 Barcelona, Spain; cvalverde@vhio.net; 12Medical Oncology Department, Hospital Universitari Son Espases, 07120 Palma, Spain; pablo.luna@ssib.es; 13Medical Oncology Department, Hospital Universitario Ramón y Cajal, 28034 Madrid, Spain; mariaangeles.vaz@salud.madrid.org; 14Medical Oncology Department, Hospital Universitario Central de Asturias, 33011 Oviedo, Spain; pilarblayalbors@gmail.com; 15Medical Oncology Department, Hospital Universitario Gregorio Marañón, 28007 Madrid, Spain; rosa.alvarez.al@gmail.com; 16Medical Oncology Department, Hospital de la Santa Creu i Sant Pau, 08041 Barcelona, Spain

**Keywords:** osteosarcoma, neoadjuvant chemotherapy, pharmacogenomics, personalized medicine

## Abstract

**Background:** Osteosarcoma is a rare disease, but it is the most frequent malignant bone tumor. Primary treatment consists of preoperative MAP (methotrexate (MTX), doxorubicin and cisplatin) chemotherapy followed by surgery and adjuvant chemotherapy. Pathological response to preoperative chemotherapy is one of the most important prognostic factors, but molecular biomarkers are lacking. Additionally, chemotherapy-induced toxicity might jeopardize treatment completion. We evaluated variants in genes involved in DNA repair and drug metabolism pathways as predictors of response to MAP-based treatment. **Material and Methods:** Germline polymorphisms in *MTHFR*, *SLC19A1*, *ABCB1*, *ABCC2*, *ABCC3*, *ERCC1*, *ERCC2* and *GSTP1* genes were determined for association studies in 69 patients diagnosed with localized osteosarcoma who enrolled in the prospective GEIS-33 trial. P-glycoprotein expression in tumor tissue was also analyzed. **Results:** In the multivariate analysis, the *ABCC2* rs2273697 (odds ratio [OR] 12.3, 95% CI 2.3–66.2; *p* = 0.003) and *ERCC2* rs1799793 (OR 9.6, 95% CI 2.1–43.2; *p* = 0.003) variants were associated with poor pathological response. P-glycoprotein expression did not correlate with pathological response. The *ABCB1* rs1128503 (OR 11.4, 95% CI 2.2–58.0; *p* = 0.003) and *ABCC3* rs4793665 (OR 12.0, 95% CI 2.1–70.2; *p* = 0.006) variants were associated with MTX grade 3–4 hepatotoxicity. **Conclusions:** Our findings add to the evidence that genetic variants in the ABC transporters and DNA-repair genes may serve as predictive biomarkers for MAP chemotherapy and contribute to treatment personalization.

## 1. Introduction

Standard first-line treatment for localized high-grade osteosarcoma consist of neoadjuvant chemotherapy based on high-dose methotrexate (MTX), doxorubicin and cisplatin (the so-called MAP regimen), followed by complete surgical resection of the primary tumor and subsequent adjuvant chemotherapy [1,2,3]. However, despite multimodality treatment, the 5-year survival rate is around 70%. Risk stratification of patients is based on pathological response to neoadjuvant MAP chemotherapy that correlates with prognosis [4,5]. Patients with a pathological response ≥90% are considered good responders. However, more than 40% of the patients do not achieve a good response [6]. Strategies to improve survival in poor responders, such as the addition of chemotherapy agents such as ifosfamide plus etoposide to adjuvant chemotherapy, have been evaluated in a large-scale clinical trial, but with negative results [7,8]. Identification of poor responders at diagnosis could improve clinical outcomes by treatment escalation, or by introducing alternative therapeutic agents or targeted therapies to the neoadjuvant setting. Additionally, in a subset of patients, multidrug chemotherapy regimens could lead to early severe adverse effects such as hepatotoxicity and myelotoxicity [9], that cannot be anticipated due to the lack of predictive biomarkers of chemotherapy toxicity.

Cytostatic drugs used in the treatment of osteosarcoma exert their antitumoral activity by interfering with cell proliferation processes that ultimately lead to apoptosis. High-dose MTX acts through the folate cycle by affecting de novo synthesis of pyrimidine and purine nucleotides, both of which are essential for DNA and RNA syntheses [10]. Cisplatin and doxorubicin bind to DNA, inhibiting DNA replication [11]. The nucleotide excision repair (NER) pathway is involved in the removal of platinum adducts [12] and the glutathione-S-transferase (GST) enzymes in the detoxification processes [13]. The ATP-binding cassette (ABC) family includes the P-glycoprotein encoded by the *ABCB1* gene. The P-glycoprotein effluxes the chemotherapeutic agents back into the intestinal lumen, affecting their exposure and clearance.

Polymorphisms in genes involved in these processes may alter protein functionality and explain, at least partially, individual variation in response to MAP chemotherapy. In osteosarcomas, most studies investigating genetic variants as predictors of pathological response or toxicity are based on pathways related to MTX [14,15,16], cisplatin [17,18] or multidrug chemotherapy regimens [19,20,21]. Other strategies focused on testing a larger number of single-nucleotide polymorphisms (SNPs) have identified new genes associated with treatment outcomes, but the evidence of their causality remains low [22,23]. Thus, the genetic variants found to be associated in these studies need further confirmation in larger and more homogeneous cohorts before they can be used in treatment planification. Prospective pharmacogenetic studies in large clinical trials offer the perfect setting for evaluating and validating these genetic variants as predictive biomarkers of efficacy and toxicity. In the present study, we conducted a multicenter pharmacogenetic association study embedded within a prospective osteosarcoma study of the Spanish Group of Sarcoma Research (GEIS, by its Spanish acronym). Germline SNPs in genes relevant to the pharmacokinetics or pharmacodynamics of MTX, doxorubicin and cisplatin were analyzed as predictive biomarkers of response to chemotherapy in localized high-grade osteosarcomas. In the analysis, we also included P-glycoprotein expression values, determined centrally in the GEIS-33 protocol.

## 2. Materials and Methods

### 2.1. Study Design

This was a multicenter study, embedded in the GEIS-33 protocol, a prospective observational study for patients with newly diagnosed high-grade osteosarcoma localized in the extremities. Patients included in this study were enrolled in 13 tertiary hospitals in Spain.

The GEIS-33 protocol was conducted according to the provisions of the Declaration of Helsinki, and it was approved by the ethics committees of all participating centers (ISG-GEIS-OS-2). All patients or, in the case of children, their parents or guardians, provided written informed consent to participate.

### 2.2. Patients’ Characteristics

Between July 2016 and November 2020, 69 patients with non-metastatic high-grade osteosarcoma localized in the extremities, enrolled in the GEIS-33 protocol and with an available DNA sample, were included in this study. Table 1 shows the characteristics of the patients and tumors.

All patients received 2 cycles over 8 weeks of standard preoperative chemotherapy consisting of high-dose MTX 12 g/m^2^, doxorubicin 90 mg/m^2^ (Adriamycin) and cisplatin 120 mg/m^2^. After neoadjuvant chemotherapy, patients underwent surgery with curative intent. After surgery, patients were stratified according to P-glycoprotein expression and histological response. Patients with negative tumor expression of P-glycoprotein received conventional adjuvant MAP chemotherapy. Patients with positive tumor expression of P-glycoprotein were stratified according to histological response to receive mifamurtide (tumor necrosis ≥ 90%; good responders) or high-dose ifosfamide plus mifamurtide (tumor necrosis < 90%; poor responders).

### 2.3. Outcome Measures

Pathological response to neoadjuvant MAP chemotherapy was evaluated histologically in the resected surgical specimen. Pathological response classification was dichotomized into good response (tumor necrosis ≥ 90%) and poor response (tumor necrosis (<90%). Pathological response was not available for 3 patients.

Chemotherapy-induced toxicities were recorded prospectively for each drug and treatment cycle. Hepatotoxicity related to high-dose MTX treatment was evaluated based on alanine transaminase (ALT) and aspartate transaminase (AST) enzymes levels. Hematological toxicities related to doxorubicin and cisplatin were anemia, neutropenia and thrombocytopenia. Toxicities were graded according to Common Terminology Criteria for Adverse Events (CTCAE) Version 4.0. [24]. The highest grade of each toxicity was used for the analyses and dichotomized into grades 0/1–2 versus grades 3–4. Toxicity data were not available for 9 patients (n = 60), and not all toxicities were available for all 60 patients (from 58 patients for high-dose MTX hepatotoxicity and from 57 patients for hematological toxicities induced by cisplatin and/or doxorubicin).

Overall survival (OS) was calculated from the date of diagnosis to death from any cause or last clinical follow-up. Recurrence-free survival (RFS) was defined as the time from the initiation of MAP chemotherapy until the date of local or distant recurrence, whichever occurred first. Data for survival analyses were not available for all the patients (from 65 patients for OS and from 63 patients for RFS).

### 2.4. SNPs Selection and Genotyping

We selected 13 SNPs in eight genes (*MTHFR*, *SLC19A1*, *ABCB1*, *ABCC2*, *ABCC3*, *ERCC1*, *ERCC2* and *GSTP1*) related to the DNA-repair and folic acid pathways, and to the transport and detoxification of the cytostatic drugs used in high-grade osteosarcoma treatment. These polymorphisms have previously been associated with clinical outcomes in patients with osteosarcoma [14,16,20,25,26,27,28,29,30,31] (see Table 2).

DNA from peripheral blood samples was isolated by automatic extraction (Autopure, Qiagen, Hilden, Germany), and DNA concentration was measured using the NanoDrop 2000 spectrophotometer (Thermo Fisher Scientific, Wilmington, DE, USA). Samples were processed by real-time PCR using TaqMan^®^ SNP genotyping assays on a 7900 HT Real-Time PCR System (Applied Biosystems, Foster City, CA, USA). All the methods were performed following the manufacturers’ recommendations. The call rates were higher than 99% for SNPs and samples. The allele frequencies of the SNPs did not differ from those reported in European populations [32]. All the SNPs were in Hardy–Weinberg equilibrium (*p* > 0.05).

### 2.5. Immunohistochemical Studies

P-glycoprotein expression was evaluated in the biopsy at diagnosis and was prospectively recorded. P-glycoprotein determination was performed according to the GEIS-33 protocol and centralized at the Instituto Ortopedico Rizzoli (IRCCS), Italy. In brief, immunohistochemistry was performed in 4 to 6 μm thick formalin-fixed paraffin-embedded (FFPE) tissue sections using an avidin–biotin peroxidase complex method (Vectastain ABC kit; Vector Laboratories, Inc, Burlingame, CA, USA) and three monoclonal antibodies (JSB-1 (Monosan Sanbio, Uden, The Netherlands), MRK16 (MyBioSource Aurogene Srl, Rome, Italy) and C494 (Invitrogen, Ltd., Paisley, UK)) [33]. P-glycoprotein expression was classified as positive, negative or not evaluable.

### 2.6. Statistical Analysis

The study sample size had more than 80% statistical power with two-sided 95% confidence intervals (CIs) to detect genetic effect sizes of moderate magnitude (odds ratios (OR) ≤ 3), assuming 40% of poor responders to neoadjuvant MAP chemotherapy (calculated by Epi Info 7TM version 7.2.5.0 (https://www.cdc.gov/epiinfo); accessed on 9 January 2024). Chi-square was used to detect statistical differences between categorical variables. Logistic regression analyses were performed including as covariates age (4–10 years versus 10–32 years), gender and tumor site (femur/humerus versus other) for pathological response, and age and gender for toxicity. Kaplan–Meier curves and a log-rank test were used for OS and RFS analyses. Significant associations were presented with the ORs and 95% CIs. Statistical significance was defined as a *p* value < 0.05. Bonferroni correction for multiple comparisons was set at *p* < 0.001. Statistical analyses were performed using IBM SPSS Statistics version 26.0, and the statistical package PLINK version 1.07.2 [34].

## 3. Results

### 3.1. Clinical Results

The median follow-up was 62.4 (interquartile range [IQR], 38.4–80.3] months, and the median age at diagnosis was 14 (IQR, 4–32) years. Twenty-six (37.7%) patients achieved a good pathological response (≥90%) after two cycles of neoadjuvant MAP chemotherapy. During the study follow up, the disease progressed in 16 (23.9%) patients and 14 (21.2%) patients died. None of the clinicopathological variables analyzed regarding the pathological response showed statistical significance: age (*p* = 0.08), gender (*p* = 0.42) and tumor site (*p* = 0.33).

### 3.2. P-Glycoprotein Expression and ABCB1 Genetic Variants

P-glycoprotein expression in tumor samples was positive in 35 (50.7%) patients, negative in 28 (40.6%) patients and not evaluable in 6 (8.7%) patients. We analyzed the correlation between P-glycoprotein expression and the rs1045642, rs2032582 and rs1128503 *ABCB1* genetic variants determined in germline DNA. We found that the *ABCB1* rs1045642-A allele was marginally correlated with positive P-glycoprotein expression (*p* = 0.049): 51% of tumors had positive expression and 34% of tumors had negative expression (Figure 1). P-glycoprotein expression was not correlated with the other two genetic variants, rs2032582 (*p* = 0.28) or rs1128503 (*p* = 0.24).

### 3.3. Association Analyses and Pathological Response

P-glycoprotein expression was not associated with pathological response (*p* = 0.52). Table 3 shows univariate analyses for genetic variants and pathological response. Univariate analyses showed marginal associations between the *ABCC2* rs2273697 and *ERCC2* rs1799793 variants and pathological response. Patients carrying the *ABCC2* rs2273697-A allele (*p* = 0.04) and patients carrying the *ERCC2* rs1799793-A allele (*p* = 0.047) had a higher risk of poor pathological response. For *ABCC2* rs2273697, 50% of patients with the GG genotype presented poor pathological response, compared to 81.8% of patients with GA or AA genotypes (*p* = 0.02 in a dominant model). For *ERCC2* rs1799793, 45.2% of patients with the GG genotype presented poor pathological response, compared to 74.3% of patients with the GA or AA genotypes (*p* = 0.02 in a dominant model). Multivariate analyses including these two SNPs and age, gender and tumor site as covariates showed significant associations for both genetic variants: *ABCC2* rs2273697 (OR 12.3, 95% CI 2.3–66.2; *p* = 0.003) and *ERCC2* rs1799793 (OR 9.6, 95% CI 2.1–43.2; *p* = 0.003). However, these associations were not statistically significant after the Bonferroni test.

### 3.4. Association Analyses and Toxicity

Table 4 shows univariate analyses for high-dose MTX hepatotoxicity and SNPs in genes related to the folic acid pathway and drug transport. Univariate analyses showed statistically significant associations between the *ABCB1* rs1128503 and *ABCC3* rs4793665 variants and severe hepatotoxicity. Patients carrying the *ABCB1* rs1128503-G allele (*p* = 0.009) and patients carrying the *ABCC3* rs4793665-T allele (*p* = 0.04) presented a higher risk of developing grade 3–4 hepatotoxicity because of MTX treatment. For *ABCB1* rs1128503, 86.4% of patients with the GG genotype developed grade 3–4 hepatotoxicity, compared to 44.4% of patients with GA or AA genotypes (*p* = 0.002 in a dominant model). For *ABCC3* rs4793665, 90.5% of patients with the TT genotype developed grade 3–4 hepatotoxicity, compared to 43.2% of patients with TC or CC genotypes (*p* < 0.001 in a dominant model). Multivariate analyses including these two SNPs and age and gender as covariates showed significant associations for *ABCB1* rs1128503 (OR 11.4, 95% CI 2.2–58.0; *p* = 0.003) and *ABCC3* rs4793665 (OR 12.0, 95% CI 2.1–70.2; *p* = 0.006) variants. However, these associations were not statistically significant after the Bonferroni test.

The effect of a possible gene–gene interaction between these two genetic variants on MTX-induced hepatotoxicity was also investigated considering age and gender as covariates. Analyses showed statistically significant interactions between the *ABCB1* rs1128503 and *ABCC3* rs4793665 polymorphisms (*p* = 0.01).

Univariate analyses for hematological toxicities induced by cisplatin and doxorubicin treatment and SNPs in genes related to the DNA-repair pathway and drug transport showed non-significant associations (Appendix A).

### 3.5. Survival Analyses

Univariate analyses for the *ABCC2* rs2273697 and *ERCC2* rs1799793 variants associated with pathological response, and for the *ABCB1* rs1128503 and *ABCC3* rs4793665 variants associated with MTX hepatotoxicity, showed no statistically significant associations with OS and RFS (Appendix A).

## 4. Discussion

One of the most important prognostic criteria in high-grade osteosarcoma is the assessment of the pathological response, but validated molecular biomarkers for treatment stratification at the time of diagnosis are lacking. In addition, there are as of yet no molecular biomarkers to predict treatment-derived toxicity that may delay treatment or lead to serious clinical complications. In this study, we found that the *ABCC2* rs2273697 and *ERCC2* rs1799793 germline variants were associated with poor pathological response. Moreover, we found significant associations of the *ABCB1* rs1128503 and *ABCC3* rs4793665 variants with MTX-induced hepatotoxicity.

As optimization of risk-stratification is a challenge in the systemic treatment strategy in non-metastatic osteosarcoma, there is a need to identify new biomarkers. The *ABCB1* gene encodes P-glycoprotein, an efflux transporter involved in the reduction in the intracellular concentration of many toxic compounds. Overexpression of this protein in tumor tissue has been associated with a worse response to MAP chemotherapy in patients with osteosarcoma [35]. Based on this observation, a prospective trial that stratified patients according to P-glycoprotein expression was conducted in Italy (ISG/OS-2) and Spain (GEIS-33). However, the Italian group [36] reported that P-glycoprotein expression was not a predictor of pathological response in induction chemotherapy. Here, we analyzed the effect of P-glycoprotein expression in tumor tissue on pathological response and also found no association. This finding is in line with a previous retrospective study that had the limitation of small sample size [37], but also with a prospective study conducted in 685 patients with localized high-grade osteosarcoma [38]. However, it should be noted that the appropriate assessment of P-glycoprotein expression may depend on tumor heterogeneity, which could influence the observed results by underestimating possible subclonal expression of P-glycoprotein.

We also analyzed whether there was a correlation between germline variants in the *ABCB1* gene and P-glycoprotein expression in the tumor, with marginal results. This observation suggests that tumor protein expression would be a more informative biomarker than genetic variants for those proteins whose regulation can be modified by the tumor microenvironment.

Other research has focused on the characterization of genetic biomarkers that may be useful in therapeutic guidance in osteosarcoma. Most of these are association studies that have analyzed candidate SNPs for certain pathways related to MAP chemotherapy [14,15,16,17,18,19,20,21]. The significant genetic alterations revealed in these studies may have an impact on the efficacy and safety of MAP chemotherapy, but more evidence is needed before their clinical use can be considered.

In our study, we found significant associations between the *ABCC2* rs2273697-A and *ERCC2* rs1799793-A alleles and poor pathological response to neoadjuvant chemotherapy. The ABCC2 protein mediates the efflux of xenobiotic compounds such as MTX, cisplatin and doxorubicin. In vitro studies showed that a haplotype including the A-allele of the *ABCC2* rs2273697 (p.Val417Ile) variant was associated with increased protein expression [39]. We thus speculate that it could have a negative effect on the response to MAP chemotherapy. Along similar lines, other studies analyzing the *ABCC2* rs2273697 variant and the pharmacokinetics of some drugs showed associations of the A-allele with reduced oral bioavailability of talinolol [40] and with reduced dose-normalized concentration of tacrolimus [41]. However, it is noteworthy that the studies mentioned above were performed on orally administered drugs, whereas chemotherapy is administered intravenously.

*ERCC2* rs1799793 (p.Asp312Asn) is a missense variant that may reduce the DNA repair capacity of the enzyme and enhance the cytotoxic effect of cisplatin [42]. Liu et al. [17] described that patients with osteosarcoma carrying the AA genotype for *ERCC2* rs1799793 presented better response. However, other studies that analyzed the variant in relation to the pathological response did not find significant associations [14,18,43]. These data contrast with our findings, indicating that the *ERCC2* rs1799793 variant warrants further investigation in the context of MAP chemotherapy.

Hepatotoxicity is a limiting complication of high-dose MTX treatment in patients with localized osteosarcoma, leading to dose adjustment and treatment delays. We found that the *ABCB1* rs1128503-G and *ABCC3* rs4793665-T alleles were associated with grade 3–4 hepatotoxicity. However, we did not find a gene–gene interaction between the two variants on hepatotoxicity.

ABCC3 transport protein is actively involved in the removal of MTX from the hepatocytes into the blood circulation and ABCB1 in its excretion into the bile [44]. *ABCB1* rs1128503 (p.Gly412=) is a synonymous variant encoding the amino acid glycine. It is located near amino acid residues that are critical for ATP binding and ATP hydrolysis [45]. We did not find, however, a correlation between the rs1128503 variant and protein expression in the tumor, although P-glycoprotein expression in the tumor would not be representative of its activity in the liver, as P-glycoprotein is overexpressed in numerous cancer-transformed tissues [46]. In line with our results, Hattinger et al. [21] found an association between the *ABCB1* rs1128503-G allele and hepatotoxicity, defined as transaminases grade 4 in 57 high-grade osteosarcoma patients.

*ABCC3* rs4793665 is a promoter variant that has a maximum score of 1a according to the RegulomeDB database [47], suggesting it would be a functional variant. Accordingly, the *ABCC3* rs4793665-T allele has been associated with low levels of hepatic mRNA and with reduced binding affinity of nuclear factors to this promoter region [48]. The variant was also found to be associated with MTX pharmacokinetic parameters, such as the area under the concentration–time curve and the maximum concentration [16].

Since ABCB1 and ABCC3 proteins are relevant for preserving liver integrity at high doses of MTX, these variants are promising biomarkers for MTX-induced hepatotoxicity in localized osteosarcoma patients. If validated, the determination of the *ABCB1* rs1128503 and *ABCC3* rs4793665 variants before neoadjuvant MAP chemotherapy may enhance the benefits of high-dose MTX by reducing or preventing the risk of toxicity in some patients. In vitro research is warranted to elucidate the exact mechanism involving these polymorphisms in the drug-induced hepatotoxicity.

We note that the associations observed may have been conditioned by the fact that some ABC transporters show binding and transport affinity for more than one anticancer agent used in MAP chemotherapy, so their efflux activity may not only depend on germline polymorphisms, but also on drug–drug interactions.

In addition, survival analyses were also performed to explore the long-term effect of the genetic variants found to be associated with pathological response and hepatotoxicity to MTX in this study. However, we found no significant associations, probably due to the moderate sample size.

Currently, most patients diagnosed with localized disease are treated with the classical regimen of neoadjuvant and adjuvant MAP chemotherapy as clinical trials testing new therapies have reported limited improvements in survival rates. In addition to the evaluation of emerging therapies, the incorporation of predictive biomarkers into clinical trials may contribute to the achievement of better outcomes in these patients. In this sense, our observations support the utility of pharmacogenetic markers in explaining part of the variation in response to MAP chemotherapy among patients with osteosarcoma, but there are limitations. First, although this is a national multicenter study, the sample size is small. This is because osteosarcoma is a rare disease and access to study samples of patients treated with the same regimen is limited. In addition, safety data were not available for all the patients included in the study. However, these data are valuable because, despite the multidrug combination chemotherapy regimen, toxicities were prospectively recorded for each chemotherapy cycle and were analyzed for each drug. Second, our selection of genes has focused only on those harboring SNPs previously associated with osteosarcoma treatment in an attempt to validate them in a homogenous prospective cohort. Although there is currently no strong evidence of their contribution to osteosarcomas treatment, other candidate genes such as *DHFR*, *ABCC1* or *ABCG2* should be investigated in future studies with large samples. Third, additional studies on the effect of the *ABCC2* rs2273697 and *ERCC2* rs1799793 variants on protein expression in the tumor may contribute further evidence on the clinical utility of these variants as treatment predictors. Finally, this study should be considered exploratory, as none of the associations described survived Bonferroni corrections for multiple comparisons. Bonferroni corrections are, however, too conservative in candidate gene studies due to the high correlation between SNPs in the same chromosomal region.

## 5. Conclusions

Our prospective pharmacogenetic study conducted in patients diagnosed with non-metastatic high-grade osteosarcoma of the extremities and treated with neoadjuvant MAP chemotherapy shows variants in ABC transporter genes that may identify patients with poor response and patients at risk of hepatic toxicity at diagnosis. These genetic variants may help personalize treatment and select a more effective and safer neoadjuvant therapy in localized osteosarcoma. Additional validation in clinical trials will be required before these genetic variants can be incorporated into clinical practice.

## Figures and Tables

**Figure 1 pharmaceutics-16-01585-f001:**
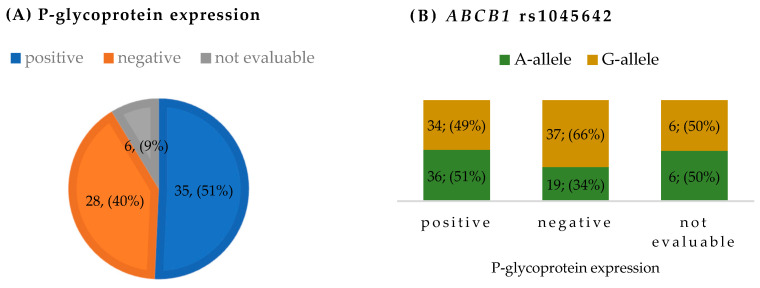
(**A**) P-glycoprotein expression in tumor samples (number of patients; %). (**B**) Distribution of *ABCB1* rs1045642 alleles according to P-glycoprotein expression (number of alleles; %).

**Table 1 pharmaceutics-16-01585-t001:** Clinical and pathological characteristics of osteosarcoma patients.

Characteristic	n (%)
Age at diagnosis (years), median (range)	14 (4–32)
Sex	
Female	32 (46.4)
Male	37 (53.6)
Primary tumor site	
Femur	45 (65.2)
Tibia/fibula	12 (17.4)
Humerus/radius	8 (11.6)
Other	4 (5.8)
Surgical margins	
Wide	44 (63.8)
Radical	7 (10.1)
Marginal	15 (21.7)
Not available	3 (4.3)
Pathological response	
≥90	26 (37.7)
<90	40 (58)
Not available	3 (4.3)
Death (Yes)	14 (21.2)
Progression (Yes)	16 (23.9)

**Table 2 pharmaceutics-16-01585-t002:** Main characteristics of the genetic variants analyzed.

Gene Symbol	Reference SNP	Molecular Consequence	Identifiers
*ERCC1*	rs11615	Synonymous	NM_001983.4:c.354T>C (p.Asn118=)
*ERCC2*	rs13181	Missense	NM_000400.4:c.2251A>C (p.Lys751Gln)
*ERCC2*	rs1799793	Missense	NM_000400.4:c.934G>A (p.Asp312Asn)
*GSTP1*	rs1695	Missense	NM_000852.4:c.313A>G (p.Ile105Val)
*ABCC3*	rs4793665	2KB upstream	NC_000017.11:g.50634726C>T
*ABCB1*	rs1045642	Synonymous	NM_001348944.2:c.3435T>C (p.Ile1145=)
*ABCB1*	rs2032582	Missense	NM_001348944.2:c.2677T>G (p.Ser893Ala); c.2677T>A (p.Ser893Thr)
*ABCB1*	rs1128503	Synonymous	NM_001348944.2:c.1236T>C (p.Gly412=)
*ABCC2*	rs2273697	Missense	NM_000392.5:c.1249G>A (p.Val417Ile)
*ABCC2*	rs3740066	synonymous	NM_000392.5:c.3972C>T (p.Ile1324=)
*MTHFR*	rs1801133	Missense	NM_005957.4:c.665C>T (p.Ala222Val) ^a^
*MTHFR*	rs1801131	Missense	NM_005957.4:c.1286A>C (p.Glu429Ala) ^a^
*SLC19A1*	rs1051266	Missense	NM_194255.4(SLC19A1):c.80A>G (p.His27Arg)

^a^ Also reported in the bibliography as: c.1298A > C (rs1801131) and c.677C > T (rs1801133); SNP: single-nucleotide polymorphism.

**Table 3 pharmaceutics-16-01585-t003:** Non-corrected univariate analyses for genetic variants and pathological response.

Pathological Response (n = 66)					
Gene	Allele	Poor Responders Frequency	Good Responders Frequency	*p*-Value	OR
*MTHFR*	rs1801131-G	0.35	0.33	0.78	1.11
*MTHFR*	rs1801133-A	0.31	0.33	0.86	0.94
*SLC19A1*	rs1051266-A	0.43	0.48	0.53	0.80
*ABCB1*	rs1045642-A	0.48	0.37	0.21	1.57
*ABCB1*	rs2032582-A	0.41	0.25	0.06	2.11
*ABCB1*	rs1128503-A	0.4	0.33	0.40	1.37
*ABCC2*	rs2273697-A	0.24	0.1	**0.04**	2.93
*ABCC2*	rs3740066-T	0.5	0.35	0.08	1.89
*ABCC3*	rs4793665-C	0.45	0.33	0.16	1.68
*ERCC1*	rs11615-G	0.45	0.33	0.16	1.68
*ERCC2*	rs13181-G	0.39	0.31	0.35	1.42
*ERCC2*	rs1799793-A	0.39	0.21	**0.047**	2.24
*GSTP1*	rs1695-G	0.43	0.48	0.53	0.80

Statistically significant *p*-values are marked in bold. None of the associations remained statistically significant after Bonferroni correction for multiple comparisons (*p* < 0.001).

**Table 4 pharmaceutics-16-01585-t004:** Non-corrected univariate analyses for high-dose MTX hepatotoxicity and variants in genes related to the folic acid pathway and drug transport.

High-Dose Methotrexate Hepatotoxicity (n = 58)					
Gene	Allele	Grade 3–4 Frequency	Grade 0–2 Frequency	*p*-Value	OR
*MTHFR*	rs1801131-G	0.43	0.26	0.07	2.13
*MTHFR*	rs1801133-A	0.24	0.41	0.05	0.46
*SLC19A1*	rs1051266-A	0.46	0.41	0.64	1.2
*ABCB1*	rs1045642-A	0.44	0.46	0.88	0.95
*ABCB1*	rs2032582-A	0.33	0.43	0.25	0.64
*ABCB1*	rs1128503-G	0.7	0.46	**0.009**	2.78
*ABCC2*	rs2273697-A	0.21	0.13	0.25	1.82
*ABCC2*	rs3740066-T	0.41	0.43	0.83	0.92
*ABCC3*	rs4793665-T	0.69	0.5	**0.04**	2.17

Statistically significant *p*-values are marked in bold. None of the associations remained statistically significant after Bonferroni correction for multiple comparisons (*p* < 0.001).

## Data Availability

The data presented in this study are not publicly available due to ethical committee regulations but are available upon request from the corresponding authors.

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
