# Peer review of "Pharmacogenetics of Neoadjuvant MAP Chemotherapy in Localized Osteosarcoma: A Study Based on Data from the GEIS-33 Protocol"

_pharmaceutics, 2024, doi:10.3390/pharmaceutics16121585_

Round 1

Reviewer 1 Report

Comments and Suggestions for Authors

The authors present a well conducted study, investigating the association of certain SNPs in drug disposition and mode of action genes with pathological response or toxicity of MAP protocol against osteosarcoma.

Overall, manuscript is well written, methods are adequately described, and the results are properly presented and discussed. Two minor aspects should be adressed:

1. The association with POOR response should be indicated in the abstract (not only saying association).

2. The authors should discuss and reference that germline findings may contradict the tumor biology. Accordingly, evaluating tumor SNPs would habe been informative.

Reviewer 2 Report

Comments and Suggestions for Authors

The study presented in the manuscript by Jaliana Salazar is well-written and structured, focusing on the identification of predictive biomarkers for neoadjuvant chemotherapy in osteosarcoma patients. However, a few suggestions for improvement can be considered:

  1. It would be valuable to include an association analysis between the complete set of SNPs and the pathological response. This could provide a more comprehensive understanding of genetic contributions to treatment efficacy.

  2. Additionally, incorporating data from Kaplan-Meier analyses to evaluate overall survival and disease-free survival would enhance the manuscript by offering insight into the prognostic significance of the identified biomarkers. This would provide a more robust picture of long-term outcomes and their potential clinical implications.

Reviewer 3 Report

Comments and Suggestions for Authors

Comments on the manuscript

The authors present their study on the role of pharmacogenetic variants on MAP chemotherapy treatment of osteosarcoma patients. The study is well-conducted, but quite small. There are some issues that need to be addressed:

1) Although the choice of genes is quite clear, it remains unclear why certain transporters or genes are not included into the analysis. Why did the authors not analyze expression or variants in DHFR as the target of MTX? DHFR is known to be upregulated in MTX-resistance. Thus, it would be interesting if there are expression changes or potential variants. MTX is also a substrate of ABCC1, while Doxo also partially by also transported by ABCG2. Why did the authors not analyze variants in these genes?

2) The authors found significant differences in ABCC2 and ERCC2 variants between the response groups. It would be interesting to show the expression of ABCC2 and ERCC2 also in the tumors, if possible (alongside P-gp expression).

4) Despite its general interest, the rationale for analysis of P-gp expression remains unclear as germline variants not necessary reflect the expression of P-gp in the tumor. Please discuss.

5) In general, a graphical representation for P-gp (results 3.2) should be added. In addition, the p-value of the P-gp expression allele is 0.049, which should be rounded to 0.05 and hence, is not significant. The way it is currently presented, is an overinterpretation of data. I suppose a graphical presentation might clarify this. Also, potentially the expression of P-gp is also subclonal and thus, overall not significant, but a resistance mechanisms in some clones? Please discuss.  

6) In the same line, please change the results for ERCC2 rs1799793 (p-value =.047).

7) The authors present their data first with non-corrected P-values showing some significances. However, after p-value correction the significances are gone. Please add the corrected p-values to the Tables and change the text accordingly. It might be misleading like this.

8) Discussion: There might also be overlapping transport affinities for the given MAP drugs. This might be the reason why there are no significant differences in the expression levels. Please add into the discussion.

Reviewer 4 Report

Comments and Suggestions for Authors

Osteosarcoma is a rare but very aggressive disease. The standard treatment approach includes preoperative chemotherapy, surgery, and postoperative chemotherapy. A drug combination is typically used as neoadjuvant therapy; however, the response in a significant proportion of patients remains poor. In this study, the investigators aimed to find biomarkers predictive of poor response and chemotherapy toxicity in osteosarcoma patients. To achieve this, a multicenter pharmacogenetic study was conducted with 69 enrolled patients. For biomarker identification, the authors analyzed 13 SNPs associated with 8 genes relevant to the pharmacokinetics or pharmacodynamics of drugs used in the preoperative stage. The authors then correlated these findings with pathological response and hepatic enzyme levels. Two SNPs were identified as correlated with pathological response, and two were associated with chemotherapy-induced toxicity. Although similar studies have been conducted and are discussed in the manuscript, this multicenter study includes a higher number of patients to help with better identification of biomarkers. The results of this study add to the current understanding of SNPs associated with treatment response and with further verification and potential validation, this would be beneficial to osteosarcoma patients.

I believe the rationale and objectives of the study are well-described, and the methodology, data analysis, presentation, and evaluation are appropriate. The authors have also effectively discussed their findings in alignment with the scope of this investigation. 

Author Response

Thank you for your encouraging comments.

Round 2

Reviewer 2 Report

Comments and Suggestions for Authors

The authors answered the reviewer's questions thoroughly and the manuscript improved significantly and is recommended for publication

Reviewer 3 Report

Comments and Suggestions for Authors

The authors addressed all my concerns.